# Mechanistic Interpretability as Statistical Estimation: A Variance Analysis of EAP-IG

## Abstract

Developing trustworthy artificial intelligence requires moving beyond black-box performance metrics toward understanding models' internal computations. Mechanistic Interpretability (MI) addresses this by identifying the algorithmic mechanisms underlying model behaviors, yet its scientific rigor critically depends on the reliability of its findings. In this work, we argue that interpretability methods such as circuit discovery should be viewed as statistical estimators, subject to questions of variance and robustness. To illustrate this statistical framing, we present a systematic stability analysis of a state-of-the-art circuit discovery method: EAP-IG. We evaluate its variance and robustness through a comprehensive suite of controlled perturbations, including input resampling, prompt paraphrasing, hyperparameter variation, and injected noise within the causal analysis itself. Across various models and tasks, our results demonstrate that EAP-IG can exhibit high structural variance and sensitivity to hyperparameters, questioning the stability of its findings. Based on these results, we offer a set of best-practice recommendations for the field, advocating for the routine reporting of stability metrics to promote a more rigorous and statistically grounded science of interpretability.

## 1 Introduction

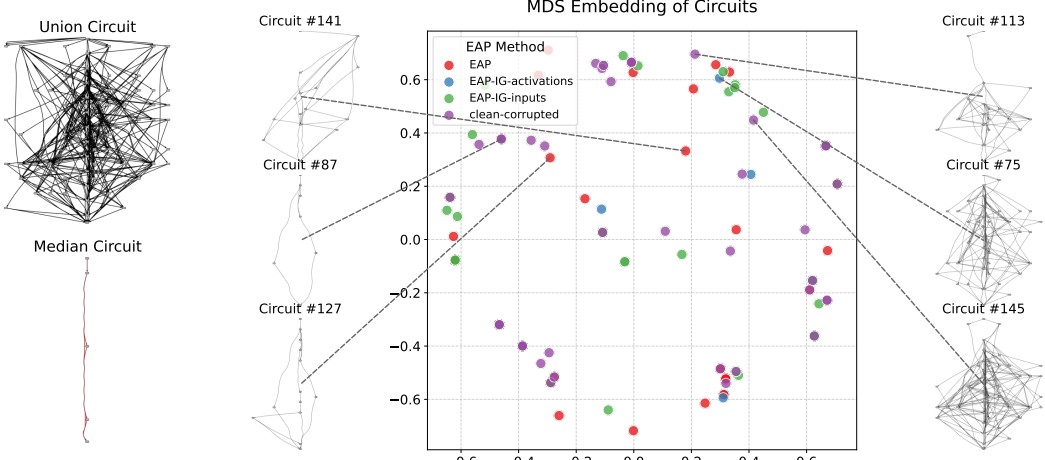

Figure 1: In gpt2-small, varying multiple circuit finding parameters at once (resampling strategy, aggregation method, type of intervention, EAP method, and pruning strategy) yields many different circuits, which we display along with the union and median circuit (left). In the center, the MDS projection of the pairwise Jaccard index matrix shows that none of the tested EAP methods consistently yields circuits with lower variance (tighter clustering).

As AI systems are increasingly deployed in real-world applications, the need for robust interpretability methods becomes more urgent. Understanding the internal mechanisms of these models is critical

not only for diagnosing failures and improving robustness (Barredo Arrieta et al., 2020), but also for complying with emerging legal frameworks that mandate explainability (Walke et al., 2025).

Mechanistic Interpretability (MI) has emerged as a promising research direction aiming to reverse-engineer the specific algorithms learned by deep neural networks (Olah et al., 2018). A central approach in MI involves identifying functional sub-networks called "circuits" that are responsible for particular capabilities (Olah et al., 2020; Elhage et al., 2021). These circuits are typically identified through interventions on the computational graph: setting the network in counterfactual states and measuring the effect of components on outputs (Vig et al., 2020a; Monea et al., 2024; Hanna et al., 2024; Syed et al., 2023b). The long-term vision of MI is to become a fully-fledged scientific discipline that studies trained models using scientific discovery tools similar to those of the natural sciences (Cammarata et al., 2020; Lindsey et al., 2025).

However, MI currently faces foundational challenges that limit its scientific rigor. Interpretability methods may produce valid explanations in random, untrained networks. For instance, feature attribution methods generate similar saliency maps for random and trained models (Adebayo et al., 2018), sparse autoencoders can extract plausible "explanations" from random weights (Heap et al., 2025), and incompatible circuits can even be discovered in networks with random behavior (Méloux et al., 2025). This highlights a non-identifiability problem: multiple incompatible explanations may satisfy current MI criteria Méloux et al. (2025), creating generalizability issues in MI explanations. For instance, circuits discovered in one setting often fail to transfer to others (Hoelscher-Obermaier et al., 2023).

To evolve from exploratory techniques into a rigorous scientific discipline, MI must adopt the standards of empirical science, most notably statistical inference (Fisher, 1955; Mayo, 1998). Scientific validity requires formulating testable hypotheses (Popper, 1934; Kabir, 2016), quantifying observational variability (fluctuations due to sampling or measurement noise; Box et al., 2005; Breznau et al., 2022), and representing uncertainty in our conclusions via measures of variability like confidence intervals (Lele, 2020; Committee et al., 2018). However, MI has yet to systematically integrate those practices. Circuits are often reported without quantification of their statistical stability, robustness to perturbations, and uncertainty estimates (Rauker et al., 2023). For instance, how does altering the dataset slightly, shifting the input distribution, or resampling change the discovered circuit? How sensitive are the findings of circuit discovery methods to hyperparameters? These limitations prevent us from assessing the generalizability, reliability, and ultimately, the validity of MI explanations Rauker et al. (2023); Liu et al. (2025); Ioannidis (2005).

In this work, we argue that mechanistic interpretability must be reframed as a problem of statistical inference. As a case study, we focus on a family of state-of-the-art MI techniques: Edge Activation Patching (EAP; Syed et al., 2023a) and its variants, notably EAP with Integrated Gradients (EAP-IG; Hanna et al., 2024). We use EAP to systematically investigate how principles of statistical robustness and variability apply to the outputs of MI methods: the discovered circuits themselves.

Our empirical analysis evaluates the stability of EAP-generated circuits under controlled variations: shifts in input distributions, bootstrap resampling of input data, and changes in method hyperparameters. We conduct experiments across three tasks and three model architectures, providing both qualitative and quantitative evidence of the circuits' variability. Our results show that the circuits identified by EAP exhibit high variance under data resampling and are sensitive to hyperparameter choices: small perturbations in data or changes in the analysis pipeline often yield substantially different circuit structures. This is visually summarized in Fig. 1, which shows the disparity of circuits found when different perturbations are applied simultaneously. In light of these findings, we propose a set of best practices for the MI community, including the systematic use of bootstrap resampling and the reporting of stability metrics to foster a more rigorous and reliable science of interpretability.

## 2 RELATED WORK

### 2.1 CIRCUIT DISCOVERY AND CAUSAL MEDIATION ANALYSIS

Numerous methodologies exist to identify the circuits central to MI's goals. Causal Mediation Analysis (CMA) (Pearl, 2001; VanderWeele, 2016) provides a formal framework that investigates how an intervention (e.g., an input) affects an outcome (e.g., a model's prediction) via mediators (e.g.,

neuron activations). In deep neural networks (DNNs), CMA helps test hypotheses about internal components' causal roles. Interventional techniques like activation patching (Vig et al., 2020b; Geiger et al., 2021; Hanna et al., 2024) manipulate mediators to quantify their influence.

Building on CMA, circuit discovery methods have evolved from feature visualization (Zeiler & Fergus, 2014; Sundararajan et al., 2017) to techniques identifying interconnected structures. Notable examples include causal tracing (Meng et al., 2022) and its variants (Meng et al., 2023; Fang et al., 2025), as well as methods like Automated Circuit Discovery (ACDC; Conmy et al., 2023, which employs activation patching to find interpretable circuits. Other lines of inquiry explore program synthesis via MI, though applications have focused on simpler architectures such as RNNs (Michaud et al., 2024). Our work focuses on Edge Activation Patching with Integrated Gradients (EAP-IG) (Hanna et al., 2024). EAP combines causal patching with gradient-based attribution (integrated gradients for EAP-IG) to score individual edge importance, and also measures the impact of excluded components when producing a circuit. We selected the EAP family for its reported state-of-the-art performance in identifying sparse, fine-grained edge-level circuits (Syed et al., 2023a; Hanna et al., 2024), making it an ideal candidate for studying the stability of such granular discoveries.

## 2.2 Evaluation of Circuit Discovery Methods

A core challenge in MI is the absence of "ground truth" circuits, as the notion of a single correct circuit can be ill-defined or non-identifiable (Mueller et al., 2024; Méloux et al., 2025). Thus, evaluation relies on proxy metrics assessing desirable properties: **faithfulness** (how accurately the circuit reflects model behavior, often tested by perturbing or ablating the identified circuit components within the full model; Conmy et al., 2023; Hedström et al., 2023; Hanna et al., 2024; Shi et al., 2024b), **sufficiency/predictive power** (whether the isolated circuit can reproduce the target behavior; Bau et al., 2017; Yu et al., 2024; Shi et al., 2024a), **interpretability** (a qualitative assessment of understandability and alignment with intuition; Olah et al., 2020), and **sparsity/minimality** (a preference for simpler, concise circuits; Elhage et al., 2021; Hedström et al., 2023; Dunefsky et al., 2024; Shi et al., 2024a). These criteria are often applied post-hoc and qualitatively. For example, Shi et al. (2024a) provides hypothesis tests for the faithfulness of a single fixed circuit, while our framework evaluates the statistical variance of the estimation process itself.

## 2.3 Stability and Robustness in Circuit Discovery

Robustness challenges have been discussed in MI methods. For example, interventions based on discovered circuits may not generalize reliably; edits derived from methods like causal tracing can fail to extend to novel contexts, casting doubts on the robustness of the underlying identified mechanism itself (Hoelscher-Obermaier et al., 2023). Furthermore, MI outputs can be prone to "interpretability illusions", where analytical techniques might highlight artifacts to statistical correlations rather than genuine computational mechanisms (Lange et al., 2023). The challenge of non-identifiability, where multiple distinct and incompatible circuits can equally satisfy common evaluation metrics (Méloux et al., 2025), further complicates claims about discovering *the* true underlying circuits.

Existing studies have also explored the sensitivity of circuit metrics to specific intervention strategies (Miller et al., 2024; Bhaskar et al., 2024). In addition, activation patching strategies may yield inconsistent results (Zhang & Nanda, 2024). To our knowledge, dedicated studies analyzing the stability of circuit discovery outputs to variations in input data, experimental conditions, or method hyperparameters are scarce. This paper aims to fill this gap by empirically studying the stability of EAP-derived circuits, thereby contributing to developing more rigorous evaluation practices in MI.

## 3 Formal Setup

This section outlines a general framework for quantifying the stability of circuits discovered by MI methods, which we then apply to the EAP family as a case study. We identify two main sources of instability in discovered circuits:

- **Variance** refers to the statistical variability of the discovered circuit (i.e., the output of the MI method) when resampling the input data used for its discovery. It captures the sensitivity of the

method to the specific sample of data drawn from an underlying distribution. This aligns with standard statistical notions of sampling variance.

- **Robustness** refers to the stability of a discovered circuit when subjected to controlled changes in the analytical setup. These changes can include variations in the MI method's hyperparameters or perturbations to the experimental conditions used during circuit discovery (e.g., adding noise to interventions). This assesses the circuit's sensitivity to the researcher's methodological decisions and the specifics of the analysis pipeline.

We aim to move beyond treating discovered circuits as singular, definitive findings. Instead, in line with modern statistical thinking that cautions against over-reliance on single point estimates (Wasserstein & Lazar, 2016), we wish to provide quantitative measures of their stability and associated performance characteristics under these different sources of perturbation, thereby offering a more nuanced understanding of their reliability.

## 3.1 General Formalization of Circuit Discovery

Let $M_\theta$ be a trained neural network. A general circuit discovery process aims to identify a subgraph (circuit) $C = (V_C, E_C)$ within $M_\theta$. This process typically involves:

- Input data ($D$): A dataset of input samples $x_i$ used specifically for the circuit discovery analysis (typically distinct from the dataset originally used to train the model $M_\theta$. These inputs are chosen to elicit distinct model behaviors or internal states that the MI method will then analyze.

- Experimental conditions ($\mathcal{E}$): The strategy for causal analysis. This specifies how interventions are performed on $M_\theta$'s internal components (e.g., neuron activations, edge weights), including which components are targeted, how their states are modified (e.g., ablated, patched), and which aspects of the model's behavior (e.g. specific logits, loss changes) are measured to quantify the effects of these interventions.

- Observational data generation: The application of experimental conditions $\mathcal{E}$ to model $M_\theta$ with inputs from $D$ produces a set of observations $O$. This data $O = \text{Observe}(M_\theta, D, \mathcal{E})$ consists of quantitative measurements (e.g., changes in model loss, output probabilities, internal activation patterns) corresponding to each intervention performed.

- Component scoring and circuit identification algorithm ($\mathcal{A}$) and hyperparameters ($\Lambda$): This stage usually involves two steps: First, individual components (e.g., edges) are assigned scores based on the observation data $O$ (e.g., their estimated impact on a task metric). Then, a circuit identification algorithm selects the final circuit (subset of components) from these scores using specific selection criteria and hyperparameters (e.g., number of edges to keep, threshold, search strategy).

The discovered circuit $C$ is thus a composite output: $C = \mathcal{A}_\Lambda(\text{Observe}(M_\theta, O, D, \mathcal{E}))$. For simplicity, we represent the entire circuit discovery method as $\mathcal{F}_{CD}$, such that $C = \mathcal{F}_{CD}(M_\theta, D, \Lambda_{\text{method}})$, where $\Lambda_{\text{method}}$ collectively represents all parameters governing $\mathcal{E}$ and $\mathcal{A}$. Different MI methods make distinct choices for $D$, $\mathcal{E}$, $\mathcal{A}$, and its hyperparameters.

Our study focuses on EAP (Syed et al., 2023a) and its variants (Hanna et al., 2024). While the underlying principles of EAP can be applied to score nodes (such as neurons or attention heads), our investigation centers on its common use for identifying important edges. EAP methods first involve an edge scoring stage, where individual edges are scored based on their influence on a pre-defined task-specific performance metric or loss function when subjected to causal interventions (patching). Following edge scoring, a circuit selection stage is employed. This stage uses the computed edge scores, a chosen selection algorithm, and hyperparameters to determine the final set of edges in the circuit $C$. The EAP variants primarily define different methodologies for the edge scoring stage. They differ in their choices for $\mathcal{E}$ (specifically, how edge effects, reflected as changes in the task metric, are measured through patching and input/activation interpolation) and the initial part of $\mathcal{A}$ (how raw observational data $O$ is processed into edge scores). These scoring methods are as follows:

- Base EAP: Computes a first-order approximation of each edge's indirect effect (the estimated change in the task metric upon corrupting the edge) by multiplying the change in downstream activations $a_x$ by the gradient of the task metric with respect to $a_x$, evaluated on clean inputs.

- EAP-IG (inputs): An adaptation of EAP that improves circuit quality by averaging the gradient of the task metric (with respect to input embeddings) over $m$ interpolation steps between clean and corrupted input embeddings, then using this to estimate edge importance.

- EAP-IG (activations): Similar to EAP-IG (inputs), but estimates edge importance by averaging the gradient of the task metric w.r.t. intermediate activations while interpolating these directly between their clean and corrupted values (for nodes) or the activations influencing an edge.

- Clean-Corrupted: A simplified variant that scores components based on the change in the task metric or its gradient measured only in the clean and corrupted states.

After scoring, circuit selection can use several algorithms such as greedy search (working backward from output logits or forward from inputs), threshold pruning, or top-N pruning. These selection algorithms have their own hyperparameters, such as whether to use absolute values, the number of edges N, or the specific threshold. They may also include steps to ensure graph connectivity. In our work, we consistently follow the iterative greedy search procedure described in the original EAP-IG paper (Hanna et al., 2024). We select an initial set of $n$ edges based on the absolute values of their scores (starting with $n = 30$), then incrementally increase $n$ up to 2000 until a path from input to output is found within the selected subgraph. If this fails, we say that no faithful circuit is found.

The operational hyperparameters we investigate for the edge scoring stage of these four EAP methodologies are the type of aggregation (how multiple scores contributing to an edge's final importance are combined, e.g., mean, median) and intervention (the type of corruption applied during patching, e.g., zero ablation, patching from a corrupted input, mean or mean-positional ablation).

## 3.2 VARIANCE, ROBUSTNESS, AND CIRCUIT PROPERTIES

We evaluate the properties of each discovered circuit $C_k$ (generated under a specific condition $k$, such as a particular data sample $D_k$ or hyperparameter setting $\Lambda_k$. We generate a set of $N$ circuits $\{C_1, C_2, \ldots, C_N\}$ by varying these conditions (e.g., through bootstrapping $D$ or changing $\Lambda$, then analyze the statistics of these properties across the set.

**Circuit performance metrics**. Those assess how well each individual circuit $C_i$, when operating as a standalone model $M_{C_i}$, replicates the task-specific behavior of the original full model $M_\theta$. These metrics are evaluated on a relevant evaluation dataset $D_{\text{eval}}$. In many circuit discovery settings, including typical EAP-IG usage, $D_{\text{eval}} = D$, in order to assess the *faithfulness* of the circuit to observational evidence, while using a separate test set would assess *generalization* to unseen data. In this paper, we therefore follow the common practice where $D_{\text{eval}} = D$, and report the mean $\mu$, variance $\sigma^2$, and coefficient of variation $CV = \sigma/\mu$ of each circuit performance metric.

- **Circuit Error:** This measures the frequency with which the circuit $M_{C_i}$ produces a different prediction than the full model $M_\theta$ on $D_{\text{eval}}$. For tasks where a discrete prediction $M(x)$ can be derived from the model's output for an input $x$, circuit error is defined as $\text{CE}(C_i, M_\theta) = \frac{1}{|D_{\text{eval}}|} \sum_{x \in D_{\text{eval}}} \mathbb{1}[M_{C_i}(x) \neq M_\theta(x)]$.

- **Circuit Divergence:** The Kullback-Leibler divergence $D_{\text{KL}}(P_{M_\theta(y|x)} || P_{M_{C_i}(y|x)})$ between the full output probability distributions of $M_\theta$ and $M_{C_i}$, averaged over $D_{eval}$. This quantifies the overall difference in predictive distributions.

**Circuit structural similarity metric (Jaccard Index)**. This measures the consistency of the *structure* (edges/nodes) of the discovered circuits themselves, independent of their performance. For any pair of circuits $C_i, C_j$ from the $N$ discovered circuits, with respective edge sets $E_i, E_j$, the Jaccard index is $J(E_i, E_j) = \frac{|E_i \cap E_j|}{|E_i \cup E_j|}$. We report the mean and variance of the pairwise Jaccard indices.

## 3.3 ASSESSING STABILITY

We investigate the stability of discovered circuits across multiple dimensions. For each experimental run (iterated over seed values), we apply one of the following variations:

**Input data resampling (bootstrap).** To estimate the variance of circuit properties attributable to the specific input data sample $D$, we employ bootstrap resampling (Efron & Tibshirani, 1986). $n = 100$

datasets are created by resampling with replacement from the original dataset $D$ with a fixed sample size of $|D|/5$. The circuit discovery method is then applied to each resampled dataset.

**Data meta-distribution shifts.** To assess circuit stability when the input data originates from related but distinct data-generating processes, we either generate multiple independent datasets from the same underlying meta-distribution (meta-dataset) or replace input prompts with a paraphrased version (re-prompting). We then apply the circuit discovery method to each newly generated dataset.

**Experimental intervention noise.** To evaluate circuit stability when the interventions within the experimental conditions are perturbed, we introduce noise during the intervention phase. Specifically, we add noise to the embedding of the token being patched. For a given model and task, we fix a random direction for the noise and vary its amplitude. Circuits are discovered under those various amplitudes, allowing for the analysis of their stability to such perturbations in the causal analysis itself.

**Base method comparison.** The four base EAP methodologies are applied separately to a consistent input dataset and a default, fixed set of hyperparameters for the aggregation and intervention type.

**Hyperparameter sensitivity.** For a given designated EAP variant, we vary the aggregation and intervention type while fixing other hyperparameters.

## 4 EXPERIMENTAL SETUP

We re-use three tasks and three datasets from the EAP-IG paper, consisting of pairs of clean and corrupted inputs:

- In the **Indirect Object Identification** (IOI) dataset (Wang et al., 2023), clean inputs are pairs of sentences involving two proper nouns, such as "Then, Lisa and Sara went to the garden. Lisa gave a drink to". In corrupted inputs, the name in the second sentence is replaced with another random one, such as "Sara". The task consists in predicting the missing name, and model performance is evaluated by measuring the logit difference between the missing name and the corrupted one. We use the dataset from Hanna et al. (2024) and the generator from Wang et al. (2023).

- In the **Subject-Verb Agreement** dataset (Newman et al., 2021), clean and corrupted inputs are noun phrases differing only in number (e.g., "Some worker" vs. "workers"). The model must predict a verb that agrees with the subject. Performance is evaluated using the logit difference between both forms of the reference verb. We use the generator from Warstadt et al. (2020), adapted to create only pairs of the type "The [NOUN_SG]"/"The [NOUN_PL]" for ease of application to EAP. Prompt paraphrasing was not implemented for this task due to the grammar-based nature of the data generation process.

- In the **Greater-Than** dataset (Hanna et al., 2023), clean inputs are sentences such as "The plan lasted from the year 1142 to the year 11". In corrupted inputs, the start year's last two digits are replaced with "01". The model is then asked to predict a year that must fall in the correct range. The evaluation metric is the difference in probability between correct and incorrect outputs. We use the dataset from Hanna et al. (2024) and the generator from Hanna et al. (2023).

We conduct experiments across three large language models to assess the consistency of our findings:

- **gpt2-small** (Radford et al., 2019): This model was selected due to its scale and widespread use as a foundational benchmark in numerous MI studies, including the original EAP, EAP-IP, and ACDC papers.

- **Llama-3.2-1B** (AI@Meta, 2024): This larger, recent decoder-only transformer model trained on different data allows us to test the generality of circuit stability observations on a more recent architecture.

- **Llama-3.2-1B-Instruct** (AI@Meta, 2024): The instruction fine-tuned variant of the previous model, allowing us to investigate whether the fine-tuning process, which significantly alters model behavior and capabilities, also impacts the stability characteristics of discovered circuits.

## 5 RESULTS

In all our experiments, KL divergence and circuit error are highly correlated and display similar trends; we only report the latter in this section, and the former in the appendix.

### 5.1 CIRCUIT VARIANCE UNDER DATA RESAMPLING

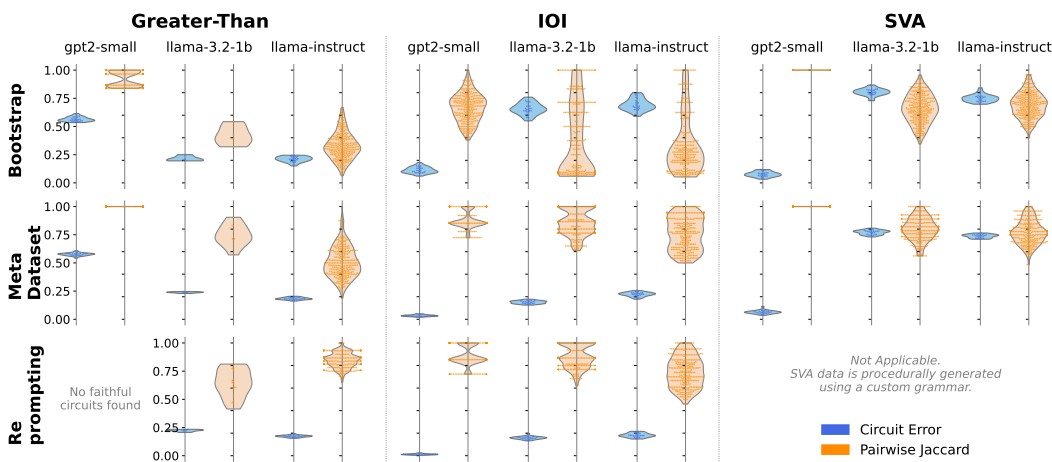

Figure 2: Circuit error and pairwise Jaccard index of EAP-IG circuits found across the three models, tasks, and types of perturbation. One point represents one circuit.

Table 1: Average value ($\mu$) and average Coefficient of Variation (CV) of the circuit error and Jaccard index across different resampling strategies, averaged over all tasks and models.

| Resampling Strategy | Circuit Error | | Jaccard Index | |
|---|---|---|---|---|
| | $\mu$ | $CV$ | $\mu$ | $CV$ |
| Bootstrap | **0.440** | 0.123 | **0.561** | **0.335** |
| Meta-Dataset | 0.300 | 0.094 | 0.790 | 0.132 |
| Prompt Paraphrasing | 0.150 | **0.134** | 0.799 | 0.131 |

We first investigate the variance of discovered circuits when the input data is resampled. Figure 2 shows the circuit error and pairwise Jaccard index for circuits discovered using different resampling strategies across all models and tasks, revealing significant variability across these axes.

We observe a notable difference in performance between GPT-2 and the larger Llama models. Circuits identified in GPT-2 consistently exhibit lower circuit error and higher structural stability (higher Jaccard index). While smaller models are often used as testbeds for developing MI methods, this suggests that circuit discovery may be more challenging and yield more unstable results in the larger, more capable models that are of ultimate interest. In contrast, we observe no notable, systematic difference between the instruction-tuned and base Llama models, suggesting that instruction tuning may not fundamentally alter the stability or discoverability of the underlying circuits (Jain et al., 2024; Prakash et al., 2024).

Furthermore, the distribution of the Jaccard index for GPT-2 appears to be multimodal, particularly visible under the bootstrap and meta-dataset resampling conditions. This suggests that the discovery process can converge to multiple, distinct, yet stable circuit solutions for the same task, echoing the concept of non-identifiability.

Finally, the choice of perturbation significantly impacts circuit stability across all models. Table 1 provides a quantitative summary of these results. Bootstrap resampling yields the highest structural variance, as indicated by the lowest average Jaccard index (0.561) and the highest CV of the Jaccard index (0.335). This suggests that circuits discovered using EAP-IG are highly sensitive to the data used in their identification. The circuits discovered under bootstrap resampling also exhibit the

highest average circuit error (0.440), indicating that the resulting circuits are not only structurally different but also less faithful to the original model's behavior. In contrast, using a meta-dataset or prompt paraphrasing results in more stable circuits, with higher Jaccard indices (resp. 0.790 and 0.799) and lower CVs. While this suggests that EAP-IG can find more consistent circuits when the data distribution is stable, even with different specific examples, the high variance under bootstrap resampling highlights a critical vulnerability of the method to sampling effects.

## 5.2 HYPERPARAMETER CHOICES AND CIRCUIT DISCOVERY

Table 2: Comparison of the circuits found in Llama-3.2-1B-Instruct on the base dataset while varying either the EAP method, aggregation or intervention performed. For each task, we report the median circuit (bold) computed across all 7 rows, as well as the Jaccard index between that median circuit and each row's circuit. Results for other models are reported in the appendix.

| Parameters | Greater-Than | | | IOI | | | SVA | | |
|---|---|---|---|---|---|---|---|---|---|
| | CErr | Size | Jacc. to Median | CErr | Size | Jacc. to Median | CErr | Size | Jacc. to Median |
| EAP, sum, patching | 0.20 | 23 | 0.417 | 0.69 | 3 | 0.286 | 0.76 | 18 | 0.536 |
| EAP-IG-activations, sum, patching | 0.20 | 17 | 0.098 | 0.69 | 12 | 0.125 | 0.76 | 24 | 0.531 |
| EAP-IG-inputs, median, patching | 0.20 | 10 | 0.086 | 0.69 | 6 | 1.000 | 0.75 | 21 | 0.840 |
| EAP-IG-inputs, sum, mean | **0.19** | **28** | **1.000** | 0.72 | 7 | 0.182 | 0.73 | 24 | 0.960 |
| EAP-IG-inputs, sum, mean-positional | 0.41 | 33 | 0.298 | **0.82** | **6** | **1.000** | 0.73 | 22 | 0.808 |
| EAP-IG-inputs, sum, patching | 0.20 | 16 | 0.571 | 0.69 | 7 | 0.182 | **0.75** | **25** | **1.000** |
| clean-corrupted, sum, patching | 0.20 | 16 | 0.419 | 0.69 | 9 | 0.071 | 0.76 | 16 | 0.577 |

We next evaluate the robustness of circuit discovery to the value of hyperparameters within the EAP-IG framework. Figure 1 (in the introduction) provides a visual summary of how varying multiple parameters at once leads to a high diversity in circuits found in gpt2-small for the IOI task.

Table 2 provides a detailed analysis for Llama-3.2-1B-Instruct across all tasks. For each task, we report the circuit error, size, and Jaccard similarity to the median circuit for different EAP variants and hyperparameters. The results show considerable variation in the discovered circuits depending on the configuration. For instance, in the Greater-Than task, the Jaccard similarity to the median circuit ranges from 0.086 to 1. Similarly, for the IOI task, some circuits have a Jaccard similarity of 1 to the median, while others are as low as 0.071. This indicates that the choice of EAP variant and hyperparameters can lead to substantially different circuits, and highlights the importance of reporting these choices and assessing their impact on the final results.

## 5.3 SENSITIVITY TO EXPERIMENTAL NOISE

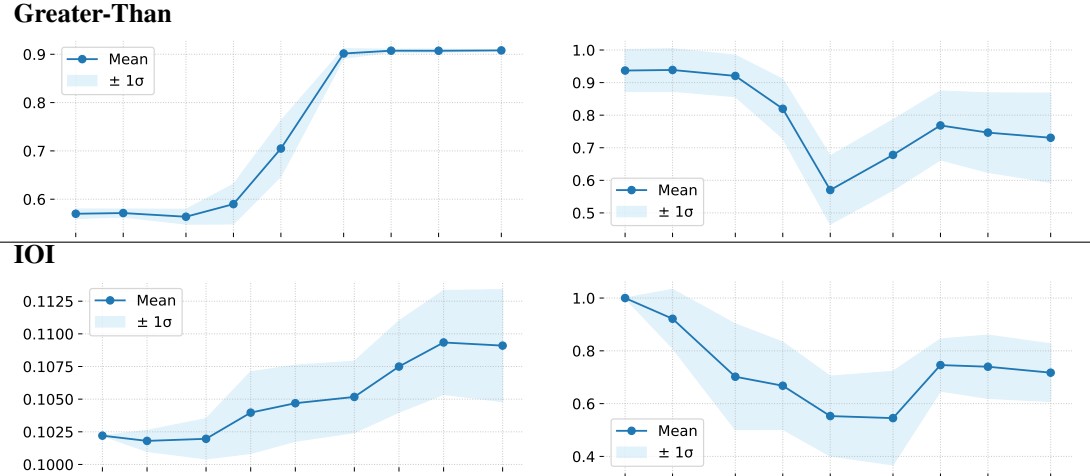

Figure 3: Average and standard deviation of the circuit error (left) and pairwise Jaccard (right) index of the circuits found in gpt2-small when using noise with amplitude [0.01, 0.02, 0.05, 0.1, 0.2, 0.5, 1, 2, 5] as intervention.

To assess the robustness of EAP-IG to perturbations in the causal analysis itself, we replace the intervention method with injected noise into the token embeddings as the intervention method. Figure 3 shows the effect of increasing noise amplitude on circuit error and pairwise Jaccard index for gpt2-small on the Greater-Than and IOI tasks.

Increasing the noise amplitude generally leads to an increase in circuit error and a decrease in the Jaccard index, indicating less stable and faithful circuits. When the noise amplitude is above 0.5, the circuits found are more stable but perform poorly. The averaged CVs of the Jaccard index and circuit error across tasks peak at a noise amplitude of around 0.2, while the average circuit error remains relatively low at this level. This suggests that a moderate amount of noise can be a useful tool for probing the stability of discovered circuits, revealing structural instabilities without drastically impacting the circuit's functional performance. A plot of the CVs can be found in the appendix.

## 6 DISCUSSION

Our empirical analysis of EAP-IG reveals significant variability in discovered circuits under perturbation, underscoring the need for a more statistically rigorous approach to MI. We find that circuits are sensitive to the specific data sample used for discovery, the choice of hyperparameters, and noise in the causal analysis. Our key findings are as follows:

- **High Variance:** Circuits discovered with EAP-IG exhibit high variance when the input data is resampled using bootstrapping. This suggests that a circuit identified from a single dataset may not be representative of the underlying mechanism and may be an artifact of the sample.

- **Hyperparameter Sensitivity:** The structure of discovered circuits is highly sensitive to the choice of EAP variant and its hyperparameters. This lack of robustness to methodological choices poses a challenge for the reproducibility and generalizability of MI findings.

- **Impact of Noise:** Adding noise to the causal interventions degrades circuit performance, but can help assess circuit stability. Moderate levels of noise can effectively reveal structural instabilities.

As a result, we propose recommendations to promote a more statistically grounded science of MI:

1. **Report Stability Metrics Routinely.** We strongly advocate for the routine reporting of stability metrics alongside circuit discovery results. Specifically, we recommend that researchers report the variance of circuit structure and performance (e.g., the average pairwise Jaccard index and the CV of the circuit error) under bootstrap resampling of the input data. This practice, common in mature scientific fields (Efron & Tibshirani, 1986; Berengut, 2006), would provide a crucial measure of the statistical reliability of the found circuits. Our publicly available codebase facilitates the computation of these metrics.

2. **Justify and Report Hyperparameter Choices.** Given the sensitivity of EAP-IG to hyperparameter settings, it is crucial that researchers transparently report and justify their choices. When possible, a sensitivity analysis should be conducted to assess the impact of different hyperparameter settings on the discovered circuits.

3. **Use Noise for Robustness Checks.** We recommend using noise injection during causal analysis as a controlled stress test for discovered circuits. Reporting how circuit stability and performance degrade with increasing noise can provide valuable insights into the robustness of the identified mechanisms. A noise level of 0.2 seems to be a good starting point for gpt2-small, as it reveals structural variance without excessively harming performance.

**Future Directions.**   Our work opens up several avenues for future research. The high variance of discovered circuits suggests that instead of seeking a single "true" circuit, it might be more fruitful to characterize a distribution over possible circuits. This could be achieved by developing methods that explicitly model the uncertainty in circuit discovery. The set of circuits generated via bootstrapping in this study is a first approximation of such a distribution.

To better diagnose the sources of instability, future work should aim to decompose the total observed variance into scoring variance (fluctuations in the attribution values, e.g., due to gradient noise) and discretization variance (instability introduced by the greedy search or thresholding steps). Distinguishing these factors is important for method development, as high scoring variance points to a

need for more robust statistical estimators (reducible error), while high discretization variance in the presence of stable scores may indicate fundamental non-identifiability (irreducible error), suggesting the existence of a set of equally valid circuits rather than a single true mechanism.

Our findings also motivate the development of new circuit discovery methods that are explicitly designed to be more robust to data sampling and hyperparameter choices. One promising direction is to incorporate a stability objective into the circuit discovery process itself, such as searching for circuits that are not only faithful but also stable across bootstrap resamples or noise perturbations.

Finally, while our study focused on EAP-IG, the statistical framework we have proposed is broadly applicable to other circuit discovery methods. We encourage the community to adopt similar stability analyses for other techniques to build a more complete picture of the reliability of MI findings. By embracing statistical rigor, we can move towards a more mature and trustworthy science of mechanistic interpretability.

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

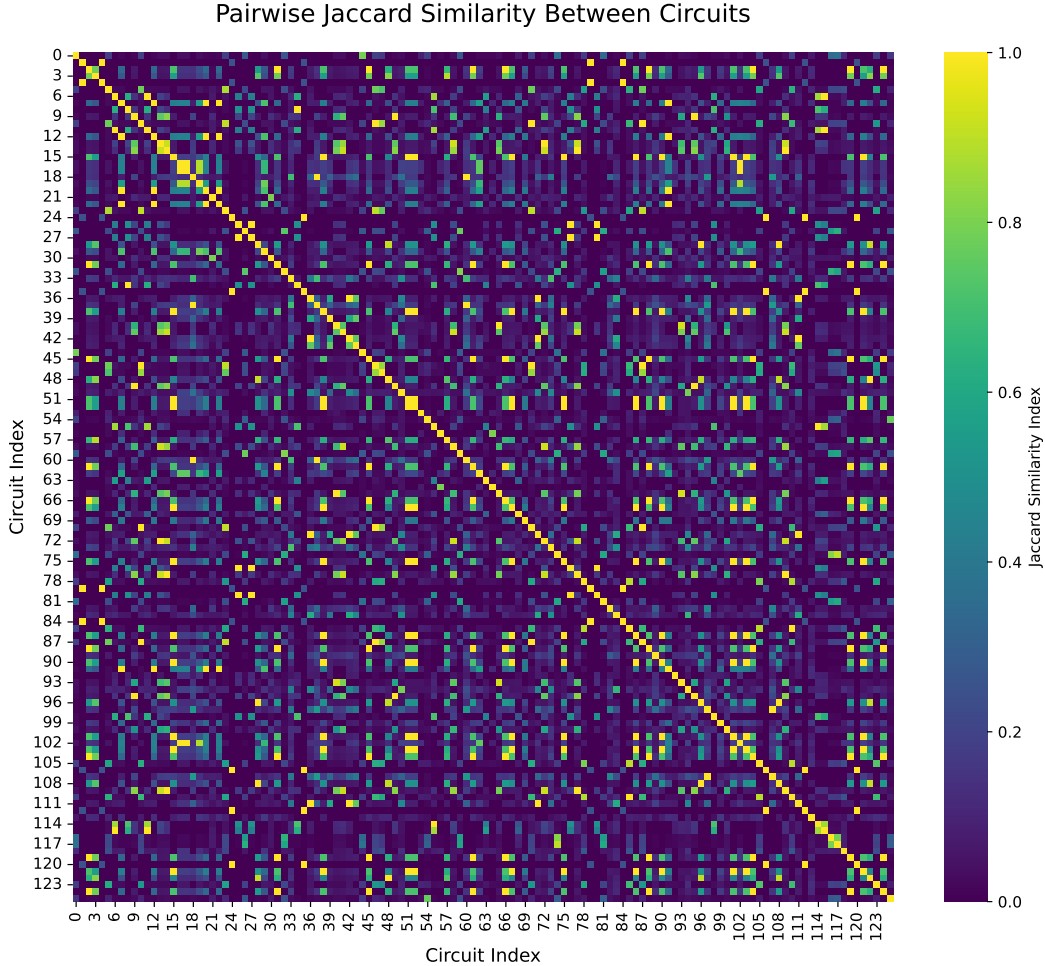

Figure 4: Full heatmap of the pairwise Jaccard index between circuits displayed in Figure 1 (circuits found in gpt2-small on the Greater-Than task while varying all parameters)

Table 3: Aggregated results from Figure 2 for bootstrap resampling.

| Model Name | Circuit Error | | | KL Divergence | | | Pairwise Jaccard Index | | |
|---|---|---|---|---|---|---|---|---|---|
| | $\mu$ | $\sigma^2$ | $CV$ | $\mu$ | $\sigma^2$ | $CV$ | $\mu$ | $\sigma^2$ | $CV$ |
| **Greater-Than** | | | | | | | | | |
| Llama-3.2-1B | 0.21 | $4.67 \cdot 10^{-4}$ | 0.10 | $6.91 \cdot 10^{-7}$ | $1.29 \cdot 10^{-14}$ | 0.16 | 0.42 | $5.93 \cdot 10^{-3}$ | 0.18 |
| Llama-3.2-1B-Instruct | 0.21 | $5.94 \cdot 10^{-4}$ | 0.12 | $6.43 \cdot 10^{-7}$ | $6.50 \cdot 10^{-16}$ | 0.04 | 0.33 | $1.36 \cdot 10^{-2}$ | 0.36 |
| **IOI** | | | | | | | | | |
| Llama-3.2-1B | 0.66 | $2.51 \cdot 10^{-3}$ | 0.08 | $5.48 \cdot 10^{-6}$ | $1.29 \cdot 10^{-13}$ | 0.07 | 0.39 | $1.07 \cdot 10^{-1}$ | 0.85 |
| Llama-3.2-1B-Instruct | 0.69 | $2.62 \cdot 10^{-3}$ | 0.07 | $9.26 \cdot 10^{-6}$ | $4.44 \cdot 10^{-13}$ | 0.07 | 0.34 | $6.72 \cdot 10^{-2}$ | 0.76 |
| gpt2-small | 0.11 | $7.32 \cdot 10^{-4}$ | 0.24 | $1.23 \cdot 10^{-6}$ | $8.80 \cdot 10^{-14}$ | 0.24 | 0.67 | $1.57 \cdot 10^{-2}$ | 0.19 |
| **SVA** | | | | | | | | | |
| Llama-3.2-1B | 0.80 | $1.02 \cdot 10^{-3}$ | 0.04 | $1.61 \cdot 10^{-5}$ | $4.02 \cdot 10^{-13}$ | 0.04 | 0.66 | $1.55 \cdot 10^{-2}$ | 0.19 |
| Llama-3.2-1B-Instruct | 0.75 | $1.04 \cdot 10^{-3}$ | 0.04 | $1.87 \cdot 10^{-5}$ | $3.97 \cdot 10^{-13}$ | 0.03 | 0.69 | $1.20 \cdot 10^{-2}$ | 0.16 |
| gpt2-small | 0.08 | $5.00 \cdot 10^{-4}$ | 0.29 | 0 | 0 | | 1.00 | 0 | 0.00 |

Table 4: Aggregated results from Figure 2 for meta-dataset resampling.

| Model Name | Circuit Error | | | KL Divergence | | | Pairwise Jaccard Index | | |
|---|---|---|---|---|---|---|---|---|---|
| | $\mu$ | $\sigma^2$ | $CV$ | $\mu$ | $\sigma^2$ | $CV$ | $\mu$ | $\sigma^2$ | $CV$ |
| **Greater-Than** | | | | | | | | | |
| Llama-3.2-1B | 0.24 | $3.06 \cdot 10^{-5}$ | 0.02 | $5.58 \cdot 10^{-7}$ | $3.56 \cdot 10^{-16}$ | 0.03 | 0.74 | $8.17 \cdot 10^{-3}$ | 0.12 |
| Llama-3.2-1B-Instruct | 0.18 | $1.05 \cdot 10^{-4}$ | 0.06 | $6.46 \cdot 10^{-7}$ | $1.31 \cdot 10^{-16}$ | 0.02 | 0.51 | $1.83 \cdot 10^{-2}$ | 0.27 |
| **IOI** | | | | | | | | | |
| Llama-3.2-1B | 0.15 | $1.67 \cdot 10^{-4}$ | 0.09 | $5.75 \cdot 10^{-7}$ | $6.68 \cdot 10^{-16}$ | 0.04 | 0.86 | $1.25 \cdot 10^{-2}$ | 0.13 |
| Llama-3.2-1B-Instruct | 0.22 | $3.30 \cdot 10^{-4}$ | 0.08 | $6.19 \cdot 10^{-7}$ | $1.53 \cdot 10^{-15}$ | 0.06 | 0.76 | $2.13 \cdot 10^{-2}$ | 0.19 |
| gpt2-small | 0.03 | $5.23 \cdot 10^{-5}$ | 0.22 | $4.72 \cdot 10^{-5}$ | $1.91 \cdot 10^{-12}$ | 0.03 | 0.88 | $5.75 \cdot 10^{-3}$ | 0.09 |
| **SVA** | | | | | | | | | |
| Llama-3.2-1B | 0.77 | $3.60 \cdot 10^{-4}$ | 0.02 | $1.54 \cdot 10^{-5}$ | $8.18 \cdot 10^{-14}$ | 0.02 | 0.80 | $1.06 \cdot 10^{-2}$ | 0.13 |
| Llama-3.2-1B-Instruct | 0.74 | $2.52 \cdot 10^{-4}$ | 0.02 | $1.84 \cdot 10^{-5}$ | $2.05 \cdot 10^{-13}$ | 0.02 | 0.77 | $1.07 \cdot 10^{-2}$ | 0.13 |
| gpt2-small | 0.06 | $2.18 \cdot 10^{-4}$ | 0.23 | 0 | 0 | | 1.00 | 0 | 0.00 |

Table 5: Aggregated results from Figure 2 for prompt paraphrasing.

| Model Name | Circuit Error | | | KL Divergence | | | Pairwise Jaccard Index | | |
|---|---|---|---|---|---|---|---|---|---|
| | $\mu$ | $\sigma^2$ | $CV$ | $\mu$ | $\sigma^2$ | $CV$ | $\mu$ | $\sigma^2$ | $CV$ |
| **Greater-Than** | | | | | | | | | |
| Llama-3.2-1B | 0.22 | $7.77 \cdot 10^{-5}$ | 0.04 | $7.09 \cdot 10^{-7}$ | $2.05 \cdot 10^{-15}$ | 0.06 | 0.64 | $1.42 \cdot 10^{-2}$ | 0.19 |
| Llama-3.2-1B-Instruct | 0.17 | $7.46 \cdot 10^{-5}$ | 0.05 | $5.43 \cdot 10^{-7}$ | $1.04 \cdot 10^{-16}$ | 0.02 | 0.85 | $4.20 \cdot 10^{-3}$ | 0.08 |
| **IOI** | | | | | | | | | |
| Llama-3.2-1B | 0.16 | $1.66 \cdot 10^{-4}$ | 0.08 | $5.42 \cdot 10^{-7}$ | $9.45 \cdot 10^{-16}$ | 0.06 | 0.88 | $1.01 \cdot 10^{-2}$ | 0.11 |
| Llama-3.2-1B-Instruct | 0.18 | $3.44 \cdot 10^{-4}$ | 0.10 | $6.06 \cdot 10^{-7}$ | $1.43 \cdot 10^{-15}$ | 0.06 | 0.74 | $1.80 \cdot 10^{-2}$ | 0.18 |
| gpt2-small | 0.01 | $2.27 \cdot 10^{-5}$ | 0.40 | $4.31 \cdot 10^{-5}$ | $1.42 \cdot 10^{-12}$ | 0.03 | 0.89 | $7.66 \cdot 10^{-3}$ | 0.10 |

Table 6: Comparison of the circuits found in Llama-3.2-1B, using a similar setup to that of Table 2.

| Parameters | Greater-Than | | | | IOI | | | | SVA | | | |
|---|---|---|---|---|---|---|---|---|---|---|---|---|
| | CErr | KL-Div | Size | Jacc. to Median | CErr | KL-Div | Size | Jacc. to Median | CErr | KL-Div | Size | Jacc. to Median |
| EAP, sum, patching | - | - | - | - | 0.64 | $5.4 \cdot 10^{-6}$ | 7 | 0.400 | 0.80 | $1.6 \cdot 10^{-5}$ | 16 | 0.355 |
| EAP-IG-activations, sum, patching | - | - | - | - | 0.64 | $5.4 \cdot 10^{-6}$ | 117 | 0.042 | 0.80 | $1.6 \cdot 10^{-5}$ | 28 | 0.421 |
| EAP-IG-inputs, median, patching | - | - | - | - | 0.65 | $5.4 \cdot 10^{-6}$ | 11 | 0.385 | 0.80 | $1.6 \cdot 10^{-5}$ | 24 | 0.923 |
| EAP-IG-inputs, sum, mean | - | - | - | - | 0.67 | $5.4 \cdot 10^{-6}$ | 5 | 0.714 | 0.75 | $1.4 \cdot 10^{-5}$ | 26 | 1.000 |
| EAP-IG-inputs, sum, mean-positional | - | - | - | - | 0.77 | $8.8 \cdot 10^{-6}$ | 8 | 0.500 | 0.69 | $1.5 \cdot 10^{-5}$ | 25 | 0.962 |
| EAP-IG-inputs, sum, patching | 0.23 | $6.0 \cdot 10^{-7}$ | 21 | - | **0.65** | $5.4 \cdot 10^{-6}$ | **7** | **1.000** | **0.80** | $1.6 \cdot 10^{-5}$ | **26** | **1.000** |
| clean-corrupted, sum, patching | - | - | - | - | 0.59 | $5.2 \cdot 10^{-6}$ | 448 | 0.016 | 0.80 | $1.6 \cdot 10^{-5}$ | 16 | 0.355 |

Table 7: Detailed results for Table 2, including KL divergence.

| Parameters | Greater-Than | | | | IOI | | | | SVA | | | |
|---|---|---|---|---|---|---|---|---|---|---|---|---|
| | CErr | KL-Div | Size | Jacc. to Median | CErr | KL-Div | Size | Jacc. to Median | CErr | KL-Div | Size | Jacc. to Median |
| EAP, sum, patching | 0.20 | $6.4 \cdot 10^{-7}$ | 23 | 0.417 | 0.69 | $9.1 \cdot 10^{-6}$ | 3 | 0.286 | 0.76 | $1.9 \cdot 10^{-5}$ | 18 | 0.536 |
| EAP-IG-activations, sum, patching | 0.20 | $6.4 \cdot 10^{-7}$ | 17 | 0.098 | 0.69 | $9.1 \cdot 10^{-6}$ | 12 | 0.125 | 0.76 | $1.9 \cdot 10^{-5}$ | 24 | 0.531 |
| EAP-IG-inputs, median, patching | 0.20 | $6.4 \cdot 10^{-7}$ | 10 | 0.086 | 0.69 | $9.1 \cdot 10^{-6}$ | 6 | 1.000 | 0.75 | $1.9 \cdot 10^{-5}$ | 21 | 0.840 |
| EAP-IG-inputs, sum, mean | **0.19** | $7.1 \cdot 10^{-7}$ | **28** | **1.000** | 0.72 | $9.3 \cdot 10^{-6}$ | 7 | 0.182 | 0.73 | $1.6 \cdot 10^{-5}$ | 24 | 0.960 |
| EAP-IG-inputs, sum, mean-positional | 0.41 | $5.7 \cdot 10^{-6}$ | 33 | 0.298 | **0.82** | $1.7 \cdot 10^{-5}$ | **6** | **1.000** | 0.73 | $1.7 \cdot 10^{-5}$ | 22 | 0.808 |
| EAP-IG-inputs, sum, patching | 0.20 | $6.4 \cdot 10^{-7}$ | 16 | 0.571 | 0.69 | $9.1 \cdot 10^{-6}$ | 7 | 0.182 | **0.75** | $1.8 \cdot 10^{-5}$ | **25** | **1.000** |
| clean-corrupted, sum, patching | 0.20 | $6.4 \cdot 10^{-7}$ | 16 | 0.419 | 0.69 | $9.1 \cdot 10^{-6}$ | 9 | 0.071 | 0.76 | $1.9 \cdot 10^{-5}$ | 16 | 0.577 |

Table 8: Comparison of the circuits found in gpt2-small, using a similar setup to that of Table 2.

| Parameters | IOI | | | | SVA | | | |
|---|---|---|---|---|---|---|---|---|
| | CErr | KL-Div | Size | Jacc. to Median | CErr | KL-Div | Size | Jacc. to Median |
| EAP, sum, patching | 0.10 | $1.2 \cdot 10^{-6}$ | 12 | 0.391 | 0.06 | 0 | 1 | 1.000 |
| EAP-IG-activations, sum, patching | 0.10 | $1.3 \cdot 10^{-6}$ | 5 | 0.042 | 0.05 | 0 | 21 | 0.000 |
| EAP-IG-inputs, median, patching | 0.11 | $1.2 \cdot 10^{-6}$ | 20 | 1.000 | 0.06 | 0 | 1 | 1.000 |
| EAP-IG-inputs, sum, mean | 0.12 | $1.3 \cdot 10^{-6}$ | 20 | 1.000 | 0.07 | $3.2 \cdot 10^{-6}$ | 1 | 1.000 |
| EAP-IG-inputs, sum, mean-positional | 0.14 | $2.1 \cdot 10^{-5}$ | 21 | 0.783 | 0.08 | $1.6 \cdot 10^{-5}$ | 1 | 1.000 |
| EAP-IG-inputs, sum, patching | **0.11** | $1.2 \cdot 10^{-6}$ | **20** | **1.000** | **0.06** | 0 | **1** | **1.000** |
| EAP-IG-inputs, sum, zero | - | - | - | - | 0.00 | 0 | 1 | 1.000 |
| clean-corrupted, sum, patching | 0.11 | $1.2 \cdot 10^{-6}$ | 19 | 0.696 | 0.06 | 0 | 1 | 1.000 |

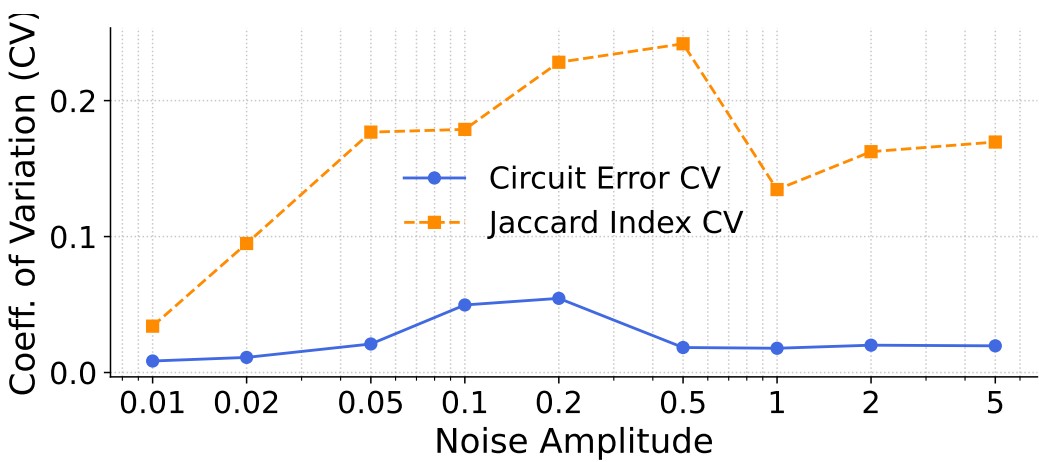

Figure 5: CV of circuit metrics for different noise amplitudes in gpt2-small, averaged across tasks.

Table 9: Detailed results for Table 3, including KL divergence. Values are plotted for noise amplitudes in [0.01, 0.02, 0.05, 0.1, 0.2, 0.5, 1, 2, 5].

| Circuit Error | KL Divergence | Pairwise Jaccard Index |
| --- | --- | --- |

**Greater-Than**

**IOI**

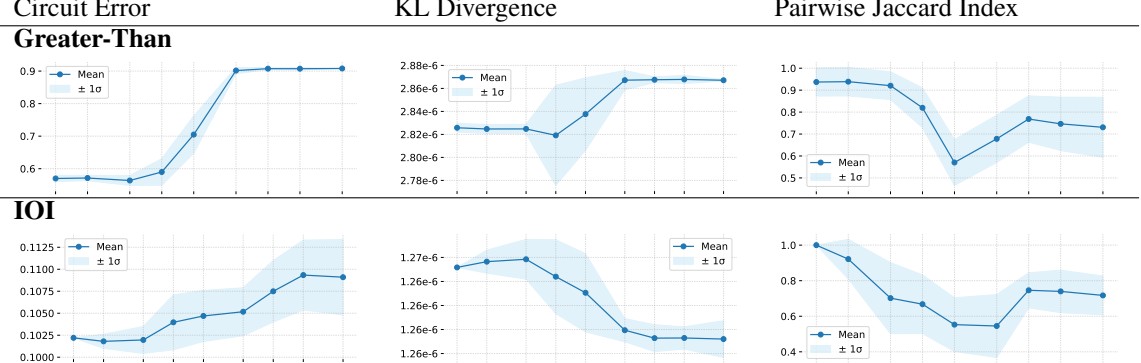

