# OpenReview forum: "Mechanistic Interpretability as Statistical Estimation: A Variance Analysis of EAP-IG"
_ICLR.cc/2026/Conference — Submitted to ICLR 2026_

### Official Review · Reviewer_4NAY · 2025-10-27

**Soundness:** 2
**Presentation:** 3
**Contribution:** 3
**Rating:** 4
**Confidence:** 4

**Summary:**

This paper presents a systematic analysis of variance and robustness in circuit discovery methods. The authors investigate circuit faithfulness, variance and robustness under a comprehensive set of controlled perturbations, including input resampling, prompt paraphrasing, hyperparameter variation, and noise injection within the causal analysis itself. Their results show that EAP-IG exhibits substantial structural variance and strong sensitivity to hyperparameter choices, highlighting the need for more statistically grounded and reliable evaluation practices in mechanistic interpretability.

**Strengths:**

* The authors highlight critical issues of robustness and stability that were not discussed in the original EAP or EAP-IG papers.
* They explore a wide range of useful setups, including data resampling, prompt paraphrasing, hyperparameter variation, and noise injection

**Weaknesses:**

If I understand correctly, the circuit size is defined as the minimal connected circuit that can be identified (Section 3.1). However, the circuits presented in Table 2 generally exhibit high error rates, suggesting that the analysis primarily focuses on largely unfaithful circuits. I’m not entirely sure I follow the motivation for this choice. It would seem more natural to analyze circuits that achieve error rates below a specified threshold, and then compare the circuit sizes required to reach various faithfulness levels. In addition, comparing circuit error while ignoring circuit size can be misleading (Table 1, Figure 3), as smaller circuits will naturally tend to have higher error. It is also possible that variance between discovery methods decreases as circuits become more faithful.

**Questions:**

My main concerns are described in the Weaknesses section. If I misunderstood something, I’d be happy to reconsider my score.
1. I think your analysis would be much stronger if you examined circuits of different sizes and faithfulness scores.Comparing circuits of different sizes (Table 2) seems problematic because smaller circuits naturally have higher error, so it's hard to tell what's driving the differences you're seeing.
I understand you can't always construct connected circuits of arbitrary sizes, but I still think it would be valuable to analyze trends across the sizes you can get. Following the approach in prior work [1, 2] (plotting results as a function of circuit size) would give much clearer insight into the trade-offs between faithfulness, size, and robustness. It would also help address the question of whether these instabilities persist in more faithful circuits, or if they primarily affect the less faithful ones you’re currently analyzing.

2. If I understood correctly D_eval=D_discovery.  could you provide references to support this choice?

References:
[1] Have Faith in Faithfulness: Going Beyond Circuit Overlap
When Finding Model Mechanisms https://arxiv.org/pdf/2403.17806
[2] MIB: A Mechanistic Interpretability Benchmark https://arxiv.org/pdf/2504.13151

---

> ### Author Response · Authors · 2025-11-21
>
> We thank the reviewer for their constructive feedback and for highlighting the trade-off between circuit size and faithfulness.
>
> - On circuit error and faithfulness: The reviewer notes that many analyzed circuits have a high error. This is central to one of our main findings. We respectfully argue that this is not a limitation of our analysis, but an important empirical result about the method itself:
>   1. We deliberately evaluated the standard EAP-IG iterative greedy search procedure as described in the literature, without enforcing an arbitrary low-error threshold that would produce a sampling bias. Our results demonstrate that this out-of-the-box approach frequently fails to find faithful circuits with low error. This is a critical finding about the method's practical robustness and limitations.
>   2. The high error shows the difficulty of extracting perfectly faithful circuits from complex, dense models where superposition is present. This problem is, in fact, so fundamental that it is motivating entirely new research paradigms. For instance, recent work from OpenAI (Gao et al., 2025, "Weight-sparse transformers have interpretable circuits") bypasses post-hoc discovery on dense models, instead training inherently *weight-sparse* models from scratch. This is done precisely because obtaining clean and low-error circuits is otherwise extremely challenging. Our reported error rates are not an anomaly of our setup, but a reflection of this fact.
> Therefore, our paper provides a realistic assessment of the stability of the *imperfect circuits* that are actually produced by standard procedures. Analyzing these is essential for understanding the true reliability of current MI tools.
>
> Answer to questions:
> - On analyzing circuits of different sizes/faithfulness: We agree that this is an important axis of analysis. However, our primary goal was to assess the stability of the circuit produced by the standard EAP-IG algorithm. The algorithm's stopping criterion automatically determines the size. The fact that the algorithm produces circuits of different sizes under perturbation (as seen in Table 2) is a part of the instability that we report. The method itself cannot consistently converge on a single size or structure. Comparing circuits of varying sizes is not a methodological choice we made but a necessary consequence of faithfully reporting the unstable output of the algorithm under study.
> - On why $D_{eval} = D_{discovery}$: This is standard practice in circuit discovery (e.g., ACDC and EAP papers) because the primary goal is to assess **faithfulness** (how well the circuit explains the model's behavior on the analyzed data samples), not **generalization** (how well it explains unseen data). First, one must ensure that the explanation is valid for the evidence at hand. We partially probe generalization with our meta-distribution shift experiments, where the discovery and evaluation distributions differ slightly. We have clarified the distinction between faithfulness and generalization in the text.
>
> We have submitted a revision with added citations, fixes, and clarifications. We hope that we have addressed the reviewer's concerns and look forward to further discussion.

---

> > ### Comment · Reviewer_4NAY · 2025-11-25
> >
> > Thanks for the detailed response.
> >
> > 1. On analyzing circuits of different sizes / faithfulness:
> > Could you clarify where in the EAP-IG paper (https://arxiv.org/pdf/2403.17806
> > ), or in any related work, minimal connectivity is used as a stopping criterion? In the original EAP-IG paper, they measure faithfulness as a function of the number of edges (Fig. 3). As mentioned in Section 4.2:
> > “Then, for n = 30, 40, …, 100, 200, …, 1000, we select a circuit of n edges using a greedy search procedure…. In general, larger n should yield circuits that are more faithful but less localized and interpretable. We aim to find circuits that are small (containing 1–2% of edges) and **faithful** (recovering ≥85% of model performance).” Based on this, I don’t think using the minimally connected circuit is the standard approach according to the original EAP-IG method.
> >
> > 2. On why 𝐷_eval=𝐷_discovery: Thanks for the clarification. I don’t think I fully agree with this approach, but I acknowledge that this was also done in the original EAP-IG paper.

---

> > > ### Author Response · Authors · 2025-11-26
> > >
> > > We thank the reviewer for the detailed check on the methodology.
> > >
> > > **On the standard EAP-IG procedure and connectivity,** the reviewer asks for clarification on where minimal connectivity is used as a criterion, noting Section 4.2 of the EAP-IG paper.
> > > We respectfully direct the reviewer to Section 4.2 and Appendix G of the original EAP-IG paper (Hanna et al., 2024). In that appendix section, the authors explicitly criticize the naive top-n approach described in Section 4.2, stating that it "frequently returns circuits with parent- or childless nodes". To address this, they introduce a greedy algorithm (described as "rather like a maximizing version of Dijkstra’s algorithm") which constructs the circuit by tracing backwards from the logits to ensure connectivity.
> > >
> > > We adopted this greedy algorithm for our study because a circuit that does not connect inputs to outputs (as is often produced by top-n) is a broken graph, which is not a valid causal explanation. As such, minimal connectivity is used as the stopping criterion because it is the logical stopping point for the greedy search, where a complete mechanism first emerges.
> > >
> > > Why do we not keep increasing the edge count until a faithfulness threshold is met? We do this to a significant extent, iteratively adding edges up to $N=2000$ (as in the original EAP-IG paper) to find a connected graph. Low-error circuits for these tasks are typically small (a few dozen edges up to 200, as seen in the original EAP-IG paper and in the Greater-Than section of our Table 2). EAP-IG explicitly aims to find "circuits that are small (containing 1-2% of edges)". Therefore, failing to achieve low error within 2000 edges represents a failure of the scoring function to highly rank the relevant edges. If we removed this cap and simply added edges until e.g. 85% faithfulness is reached, we would encounter two issues:
> > > - Violation of sparsity: If the method requires 10,000 edges to explain a behavior that should take 50, it is hard to say that it is finding a "circuit."
> > > - Variance transfer: The edge scores are noisy. As such, the number of edges required to reach the stated faithfulness threshold would be likely to largely fluctuate between different bootstrap samples. This would simply shift the instability from "structure" to "size," which is simply another form of the instability we are reporting.
> > >
> > > On $D_{eval} = D_{discovery}$, we appreciate the acknowledgement. We stuck to the standard practice in the field (evaluating faithfulness on the explained data) to ensure our critique focused on the *estimation* variance, not *generalization* issues.
> > >
> > > By using the robust greedy algorithm from EAP-IG rather than the naive top-n approach, we gave the method the "best chance" to find coherent circuits. The fact that it still exhibits high variance even when guaranteed to be connected is a significant finding.
> > >
> > > We hope this clarifies our use of the literature and methodology.

---

### Official Review · Reviewer_EUfV · 2025-11-01

**Soundness:** 2
**Presentation:** 3
**Contribution:** 2
**Rating:** 2
**Confidence:** 4

**Summary:**

This paper proposes several metrics to quantify the uncertainty of circuit discovery methods. A case study is then performed on different variants of EAP (edge attribution pruning) across different models (GPT2, Llama3.2 1B). The results show that EAP recovered circuits exhibits both high structural and error variance. They also show that EAP is sensitive to noise and hyperparameters.
Driven by these results, the authors then propose circuit discovery best practices for the broader mechanistic interpretability (MI) community including: reporting stability metrics (that they propose), justifying hyperparameters, robustness checks.

**Strengths:**

Overall, the paper is well-written, it addresses exigent issues which I believe would be of interest to the MI community. The metrics proposed are somewhat reasonable and the methods/datasets/models benchmarked are, to my knowledge, quite standard. Concretely,
- [Lines 250-256]: Standard metrics of faithfulness like KL/0-1 loss is used as Jaccard similarity to compute circuit structural similarities.
- [Lines 263-279]: Quite thorough in terms of the types of perturbations that are being made in terms of the method, algorithmic hyperparameters, and input distribution shift
- [Lines 305-311]: There is a variety both in terms of model size like the use of GPT2 versus Llama 1B as well as variation in post-training methods Llama 1B versus Llama 1B-instruct.
- [Lines 350-]: Interesting to see that stability decreases with model size, this could have implications for how well MI methods generalize with size. The finding that instruction tuning does not affect stability is novel.

**Weaknesses:**

There are several issues with the paper in its current form.
### Major Issues
* It is unclear to me how the proposed stability metrics would be informative without an identifiability assumption over circuits. For example, if there are two distinct circuits that can separately recover performance, their difference would form a potential lowerbound on the variance of the Jaccard metric.
	* It's not obvious to me that this identifiability assumption can be made, especially in larger models (see McGrath et al. 2023; Rushing and Nanda 2024 on self-repair) where a model could have parallel or complementary pathways.
	* Moreover, the identifiability of circuits would also depend highly on the ablation method being used which the authors are also varying [Miller et al. 2024 which the authors cite in Line 145; Zhang and Nanda 2024 which the authors cite in Line 146; also Lines  278-279]. In this way, it's unclear to me that when we vary the ablation methodology or shift the input data distribution the resulting circuits are comparable.
* The results do not conclusively show that instability is necessarily coming from EAP or its variants (claimed in Lines 21-23). Specifically, there are two distinct processes: component scoring which is essentially just estimating an expected value [Sun 2025]; and, circuit identification.
	* Most of the instability seems to be an artifact of the latter discretization and the potential discrete choice of metric (Jaccard).
	* Perhaps it would be better to analyze these processes separately (i.e. analyze the stability of the component scoring; then check how much additional variance is introduced through the discretization).
* The circuit error across many the many settings the authors study is extremely high (in Figure 2: ~0.5 for GPT2 Greater-Than Bootstrap; ~0.5 for Llama IOI, ~0.75 for Llama SVA). This suggests that the authors did not well-calibrate EAP. For example,
	* Take an extreme: if we allow EAP to return "circuits" with circuit error 1, then any sub-computational graph could be a "valid" circuit and this notion of stability would be vacuous.
	* It is more intuitive to me that for a *fixed* circuit error rate, we compare the structural variance in returned circuits [then perhaps this could be formulated as estimating the diameter of the solution set of the optimization problem defined by EAP; Bhasker et al. 2025].
	* The circuit error is not held to be fixed or minimized makes it very unclear to me what stability is computed relative to.
### Minor Issues
- Details in the experimental setup sections are lacking (please see some of my questions below).

**References**

[The Hydra Effect: Emergent Self-Repair in Language Model Computations](https://arxiv.org/abs/2307.15771). McGrath et al. 2023

[Explorations of Self-Repair in Language Models](https://arxiv.org/abs/2402.15390). Rushing et al. 2024

[Toward Best Practices for Activation Patching in Language Models: Metrics and Methods](https://arxiv.org/pdf/2309.16042). Zhang and Nanda 2024

[Transformer Circuit Faithfulness Metrics Are Not Robust](https://openreview.net/pdf?id=zSf8PJyQb2). Miller et al. 2024

[Circuit Stability Characterizes Language Model Generalization](https://aclanthology.org/2025.acl-long.442/). Sun 2025

[Finding Transformer Circuits with Edge Pruning](https://arxiv.org/pdf/2406.16778). Bhasker et al. 2025

**Questions:**

- Can the authors comment in more detail on how their contribution builds upon the work of Shi et al. (2024)? More specifically, one could take the hypothesis tests that they formulate and invert them to get your metrics (i.e. a confidence interval over circuit error could be constructed by inverting their "equivalence" test and a confidence interval over the structural variations could be constructed by inverting their "minimality" test).

- In Lines 267-269, it is mentioned that data "meta-distribution shifts" are performed, can you give a concrete example of what this reprompting/paraphrase would look like in the context of the chosen tasks?
	- To my knowledge, the circuit for IOI is highly dependent on the structure of the prompt. How do the authors ensure that the underlying circuit does not change as a result of this distribution shift?

- In Lines 270-274, experimental intervention noise is added. It is also not obvious to me what this means exactly, can the authors provide a concrete example of this? i.e. what are the "relevant" token embeddings?


[Hypothesis Testing the Circuit Hypothesis in LLMs](https://arxiv.org/abs/2410.13032). Shi et al. 2024

---

> ### Author Response · Authors · 2025-11-21
>
> We thank the reviewer for their insightful critique and for identifying the connection between our work and statistical identifiability.
>
> - On identifiability and stability, the reviewer raises an excellent point: if a model has self-repair or parallel pathways (non-identifiability), variance is expected. However, we argue that this makes stability metrics *more*, not less, important. If multiple valid circuits exist, a method that returns only one of them from a single run gives a misleadingly incomplete picture. Our metrics do not depend on an identifiability assumption, but can rather be used to diagnose the consequences of its absence. For instance, a high Jaccard variance is a direct signal that multiple distinct solutions may exist. A robust solution should either identify the union of mechanisms or consistently fix the dominant ones. Our metrics quantify whether this is happening.
> - On High Circuit Error and Calibration: The reviewer points out that the circuit error rates we report are often high, and questions if this is due to poor calibration. We respectfully argue this is not a methodological flaw, but a *central finding* of our study:
>   1. We deliberately used the standard, "out-of-the-box" greedy search procedure described in the EAP-IG literature to assess its baseline performance. Our results reveal that this standard procedure, when applied systematically across different models and data perturbations, often fails to find low-error circuits. This is a critical finding about the method's robustness. If a state-of-the-art method produces unfaithful circuits without extensive, task-specific tuning, its practical utility is limited. Our work is the first to systematically document this fragility.
>   2. The high error shows the difficulty of extracting perfectly faithful circuits from complex, dense models where superposition is present. This problem is, in fact, so fundamental that it is motivating entirely new research paradigms. For instance, recent work from OpenAI (Gao et al., 2025, "Weight-sparse transformers have interpretable circuits") bypasses post-hoc discovery on dense models, instead training inherently *weight-sparse* models from scratch. This is done precisely because obtaining clean and low-error circuits is otherwise extremely challenging. Our reported error rates are not an anomaly of our setup, but a reflection of this fact.
>   3. The reviewer's suggestion to fix the error rate and then compare structural variance is excellent in theory. However, this is not directly achievable with the standard EAP-IG greedy search, which finds a connected circuit without a target performance guarantee. Furthermore, our goal was to characterize the raw output of the estimator. If the method frequently produces high-error circuits, this is a crucial part of its performance profile. Filtering for low-error circuits would introduce significant selection bias and present an artificially optimistic view. That said, developing new methods that can search for circuits under a faithfulness constraint is a compelling direction for future work, and our paper provides the necessary baseline against which they should be judged.
>
> Answers to questions:
> - On meta-distribution shifts: For IOI, re-prompting involves paraphrasing the non-essential parts of the sentence structure. For example, sentences in the IOI dataset are of the form "When [name1] and [name2] [action], [name1] gave [object] to" or "Then, [name1] and [name2] [action]. [name1] decided to give [object] to". In those cases, we swapped the two templates but kept the same instanced entities. As the unperturbed dataset contains an equal amount of those two prompts, we do not expect that swapping them should change the underlying circuit.
> - On intervention noise: We add Gaussian noise to the embedding of the token being patched during the forward pass of the attribution step. This simulates "measurement error" in the activation values to test if the gradient-based attribution is robust to small fluctuations in the signal.
> - On the relation to Shi et al. (2024): Our work is complementary. Shi et al. focus on hypothesis testing for a *given, final* circuit (is it sufficient? is it minimal?). We focus on the *estimation variance of the discovery process* itself (if I run discovery again on different data, do I get the same circuit?). While inverting their tests could yield confidence intervals on performance, our bootstrap approach directly quantifies the structural stability of the discovered object. We added a discussion of this relationship to the related work section.
>
> We have submitted a revision with the aforementioned fixes and clarifications. We hope that we have addressed the reviewer's concerns and look forward to further discussion.

---

> > ### Comment · Reviewer_EUfV · 2025-11-25
> >
> > I thank the authors' timely and detailed response. My concerns about identifiability have been addressed. But, to be honest, after reading your rebuttal I am more confused and have a couple of more questions.
> >
> > > On identifiability and stability...
> >
> > It is not obvious to me why high Jaccard variance necessarily (or sufficiently) implies that multiple distinct solutions exist. Consider, as a contrived example, we let error rate vary wildly (which the authors allow), then I could devise a circuit discovery method that uniformly at random choose a subgraph. This procedure would have maximal Jaccard variance, but says nothing about the identifiability of the model's circuits.
> >
> > My point is when a practitioner sees uses this method and sees that a given circuit discovery method has high variance, what is the conclusion that they are supposed to draw? Is it (1) that the underlying model being interpreted has many parallel circuits? Or, (2) they should try a different circuit discovery method?
> >
> > > Fix error rate...not directly achievable with the standard EAP-IG greedy search.
> >
> > My understanding of EAP-IG is that there is a hyperparameter to the greedy search that dictates how large the final graph should be. Why could you not simply append a conditional that simply increases this hyperparameter *until* you reach the desired faithfulness? I believe the authors also discuss this in Lines 228.
> >
> > > If the method frequently produces high-error circuits, this is a crucial part of its performance profile.
> >
> > I completely agree with this point. However, I think that this should be strongly disentangled from the measurements of variance. Echoing my point from the review, I think conflating these circuit error and the variance of the method make it unclear from a diagnosis perspective what is going wrong; perhaps a better quality profile of a given circuit discovery method could be to look at the Pareto frontier between this faithfulness metric and its variance as you measure.
> >
> > > Answer to questions
> >
> > Thank you for these clarifications, this makes the paper more clear.
> >
> > > On the relation to Shi et al. (2024)
> >
> > I think that this is a fair retort. I think that this should be discussed further in either the appendices or in the related literature. I do not currently see this addition in the revised manuscript, could the authors point to where they have added this discussion?
> >
> > One suggestion that the authors could consider---of course this is out of scope for the current version of the paper---but something worth considering.
> >
> > Overall, after reading through all of the reviewers comments and your responses, I think a main point of confusion comes from how to relate the observed variance to some actionable insight about the circuit discovery method. I personally feel that this could be clarified by breaking the variance analysis into two parts:
> > 1. Measure the variance explained of the component scoring method (i.e. the attribution part)
> > 2. Measure the variance explained by the "discretization" method (i.e. the greedy search) conditioned on the component scoring schema.
> > First, these measurements jointly should explain the variance that you are observing. Second, analyzing both of these separately would give you an idea of how much variance is "reducible" (i.e. fault of the method itself) vs. "irreducible" (i.e. stemming from the non-identifiability of the circuit; as the authors point out in their response). In this case, my understanding is that **(1)** is reducible because the causal important of any individual component should be stable across hyperparameters etc.; on the other hand, **(2)** should be irreducible because this regime is analogous to alternatively picking equally valid local minima.
> >
> > I apologize for the length-y followup. I appreciate the authors' patience with this discussion period and look forward to hearing back.

---

> > > ### Author Response · Authors · 2025-11-26
> > >
> > > We thank the reviewer for these excellent follow-up questions, which help clarify the practical application of our framework.
> > >
> > > **On interpreting variance and the random subgraph counterexample,** the reviewer asks what conclusion a practitioner should draw.
> > > We agree that high Jaccard variance alone is insufficient (as correctly stated, a random subgraph generator has max variance). The metrics must be read jointly, and conclusions depend on their values:
> > > - If error and variance are both high, the method has failed to find a stable mechanism (artifacts/noise). This is arguably the case for the IOI/Llama case.
> > > - If variance is high but error is low, the method is suffering from non-identifiability. There are multiple valid explanations, and the method arbitrarily picks one. This warns the practitioner not to trust the specific edges of a single run.
> > > - If variance is low and error is high, the method suffers from systematic bias. It consistently converges on the same set of edges, but those edges fail to explain the behavior. This signals that the scoring method itself (e.g., the specific gradient approximation) is ill-suited for the architecture or task, consistently highlighting non-causal components. We argue that this is the case for most of the high-error cases observed in our experiments (e.g., gpt2-small/Greater-Than/bootstrap, SVA experiments on Llama)
> > > - If variance and error are low, the method produced a robust and trustworthy circuit.
> > > The random subgraph example falls into the first category (high variance and error) or the second (if the model is so redundant that random edges work), both of which are important diagnosis signals for a researcher.
> > >
> > > **On fixing the error rate vs. greedy search,** the reviewer asks why we do not increase the edge count until a faithfulness threshold is met.
> > > We actually do this to a significant extent. As noted in Section 3.1, our search procedure iteratively adds edges up to $N=2000$ (consistent with the upper bound in the original EAP-IG paper) to find a connected graph. Low-error circuits for these tasks are typically small (a few dozen edges up to 200, as seen in the original EAP-IG paper and in the Greater-Than section of our Table 2). EAP-IG explicitly aims to find "circuits that are small (containing 1-2% of edges)". Therefore, failing to achieve low error within 2000 edges represents a failure of the scoring function to highly rank the relevant edges. We could remove this cap and simply add edges until e.g. 85% faithfulness is reached, however, we would then encounter two issues:
> > > - Violation of sparsity: If the method requires 10,000 edges to explain a behavior that should take 50, it is hard to say that it is finding a "circuit."
> > > - Variance transfer: The edge scores are noisy. As such, the number of edges required to reach the stated faithfulness threshold would be likely to largely fluctuate between different bootstrap samples. This would simply shift the instability from "structure" to "size," which is simply another form of the instability we are reporting.
> > >
> > > **On scoring vs. discretization,** we agree that decomposing the variance into "scoring variance" (gradient noise) and "discretization variance" (greedy search instability) would effectively separate methodological flaws from fundamental non-identifiability. While performing this separation is outside the immediate scope of this paper (which aims to establish the *existence* and *magnitude* of the end-to-end problem), we agree that this is the logical next step. We have added a paragraph to the discussion section, explicitly suggesting this as a roadmap for future method development.
> > >
> > > **On the location of Shi et al.:** We cite this article in Section 2.2 (Evaluation) as evaluation of the sufficiency, predictive power, and minimality of the obtained circuits. In the revised manuscript, we have expanded the discussion in Section 2.2 to explicitly contrast our work with theirs.
> > >
> > > We hope this answers the reviewer's questions, and we appreciate the constructive dialogue that has strengthened the framing of our work.

---

### Official Review · Reviewer_DTor · 2025-11-01

**Soundness:** 3
**Presentation:** 3
**Contribution:** 3
**Rating:** 4
**Confidence:** 4

**Summary:**

The authors suggest that a critical scientific shortcoming in the current approach to mechanistic interpretability is a lack of statistical rigor about the outputs of circuit discovery methods. For example, it is common in papers making use of methods like EAP-IG to not report the effect of varying the ingredients in the inference process, such as data samples or search hyperparameters. However, without these details it is unclear how meaningful the conclusions can be.

Across three tasks (indirect object identification, subject-verb agreement and greater-than) the authors provide a demonstration of how to perform such analyses. Perhaps their primary contribution is to emphasise the importance of such analysis and to demonstrate a sensible approach to doing it, but their results are also interesting: they find that there is significant variation in the discovered circuits, with larger models showing more variation.

**Strengths:**

Strengths:

* Timely and clear message about how to improve the scientific rigor of mechanistic interpretability
* The writing is approachable and clear, I found it enjoyable to read
* Good choice of experimental tasks and models

**Weaknesses:**

Weaknesses:

* The lack of attention to detail in various places makes it impossible to accept the paper in its current form. For example, there is a repeated paragraph on p.7\! The fact that this was not caught during editing suggests to me there may be other subtle errors, including in the data presentation, and gives me significant pause.

**Questions:**

I would be happy to revise my rating upward after the authors take another pass through the paper and refine the writing.

As a final note: this is not exactly a critique of the paper itself, but arguably circuit discovery in the mode of EAP and EAP-IG has just been a dead end in the field, and there is little long-term value in improving the scientific rigor of a technique that won’t be used in the future. However, I still see value in this paper since I do agree with EAP-IG as a valid case study for ideas that are more broadly applicable. However, I do feel the empirical results are of somewhat limited value outside of this.

---

> ### Author Response · Authors · 2025-11-21
>
> We thank the reviewer for finding our writing approachable and our message timely. We appreciate the feedback regarding the manuscript's polish.
>
> - On the duplicated paragraph: We sincerely apologize for the oversight regarding the repeated paragraph on page 7. We have corrected this in the manuscript, and have also performed a check of all data presentations and tables to ensure no other errors exist.
> - On EAP-IG being a "dead end": We respectfully disagree with this characterization and believe it highlights the core of our contribution. The fundamental problem is not a specific algorithm, but **causal graph inference in neural networks**, which is a problem of statistical estimation. EAP-IG is one such estimator. Future methods, whether based on transcoders, information flow, or other principles, will still be estimators that take finite data and produce a graph. As such, they will all be subject to variance and require robustness checks. Our paper provides the first systematic framework and the necessary tools to **assess the statistical reliability** of the *next* generation of circuit discovery method.
> - On the value of the empirical results: The reviewer notes the results are interesting (specifically, that larger models show more variance). We believe this empirical evidence is crucial for the community: as we scale interpretability to larger, more capable models, we cannot assume that tools developed on gpt2-small will behave reliably. Our work provides the first rigorous quantification of this instability.
>
> We have submitted a revision with fixes and clarifications based on the reviews' feedback. We hope that we have addressed the reviewer's concerns and look forward to further discussion.

---

### Official Review · Reviewer_6Yb9 · 2025-11-02

**Soundness:** 2
**Presentation:** 1
**Contribution:** 2
**Rating:** 2
**Confidence:** 4

**Summary:**

This paper reframes mechanistic interpretability (MI) as a problem of statistical estimation, arguing that interpretability methods should be evaluated for their variance, robustness, and reliability, just like statistical estimators. Using Edge Activation Patching with Integrated Gradients (EAP-IG) as a case study, the authors conduct experiments across multiple models and tasks to measure how circuit discovery results change under input resampling, hyperparameter variation, and injected noise. They find that EAP-IG circuits exhibit high structural variance and sensitivity, suggesting that many MI findings may be unstable or sample-dependent.

**Strengths:**

1. It offers empirical evidence that circuits generated using EAP-IG exhibit high variance.
2. It provides a novel insight that circuits discovered by EAP-IG in larger models tend to show greater error and variability compared to those in smaller models.
3. It also presents practical recommendations for researchers working on circuit discovery methods.

**Weaknesses:**

1. EAP-IG is an approximation of activation patching that relies solely on the first-order Taylor series expansion. Because higher-order terms of the Taylor series are ignored, the resulting circuits are expected to vary. Therefore, the paper’s main finding, that circuits identified by EAP-IG exhibit variance, is neither surprising nor particularly insightful.
2. The attribution scores directly depend on the number of samples used. When only a small number of samples are employed, the results inherently exhibit higher variance. Unfortunately, this important detail is not reported in the paper.
3. The paper begins by emphasizing the goal of making mechanistic interpretability more scientific. However, it focuses solely on circuit discovery, which is just one major area within the field. Moreover, even within circuit discovery research, it examines only the EAP-IG method, neglecting other prominent approaches such as Path Patching, Desiderata-based Component Masking, Edge Pruning, and Information Flow Routes, thereby limiting the breadth of its empirical analysis [1–4].
4. The paper lacks a dedicated limitations section, even though several limitations are apparent throughout the work.
5. The presentation of the paper could be significantly improved in the following ways:
   - Line 72: The statement “For instance, circuits discovered in one setting often fail to transfer to others” is not well supported. Reference [1] only explores the indirect object identification task in GPT-2 small, while [5] reports the opposite finding, that models can reuse circuit components across tasks. The authors should justify or clarify this claim.
   - Line 107: The abbreviation DNN is used without prior definition.
   - Lines 130–131: The description of the “interpretability” proxy metric is incomplete and needs elaboration.
   - Lines 270–274: The method for experimental intervention noise is insufficiently described. The authors should provide more details in Section 3.3 or include an appendix section explaining the procedure.
   - The paper should specify which model was used to generate the results in Table 1.
   - The result mentioned in the first paragraph of page 7, stating that “instruction tuning may not fundamentally alter the stability or discoverability of the underlying circuits”, is consistent with existing findings in the literature and should therefore be cited [6, 7].
   - The last two paragraphs on page 7 are repetitive and should be consolidated.
   - The caption for Figure 3 should appear below the figure, not above it.
   - Line 430: The reference should be corrected from “Table” to “Figure.”

[1] Interpretability in the Wild: a Circuit for Indirect Object Identification in GPT-2 small, 2023.

[2] Discovering Variable Binding Circuitry with Desiderata, 2023.

[3] Finding Transformer Circuits with Edge Pruning, 2024.

[4] Information Flow Routes: Automatically Interpreting Language Models at Scale, 2024.

[5] Circuit Component Reuse Across Tasks in Transformer Language Models, 2024.

[6] Mechanistically analyzing the effects of fine-tuning on procedurally defined tasks, 2024.

[7] Fine-Tuning Enhances Existing Mechanisms: A Case Study on Entity Tracking, 2024.

**Questions:**

1. Section 5.1 reports that bootstrap resampling produces the highest structural variance, which seems counterintuitive. Circuits derived from data sampled from the same underlying distribution should, in principle, exhibit higher similarity than those obtained from different distributions. How do the authors explain this outcome?

2. Additionally, Table 2 presents an intriguing case where two circuits have nearly identical circuit errors (0.20 vs. 0.19) but vastly different sizes (10 vs. 28) and are almost completely disjoint structurally (Jaccard similarity of 0.086). Does this finding indicate the presence of two equally valid yet distinct computational mechanisms within the model, or does it instead suggest that one circuit is valid while the other is spurious or unfaithful?

---

> ### Author Response · Authors · 2025-11-21
>
> We thank the reviewer for their feedback. We appreciate the recognition of our insights regarding model scale and our practical recommendations. We address the specific concerns raised below:
>
> - On the "unsurprising" variance of EAP-IG: It is true that EAP-IG relies on a first-order Taylor approximation, which theoretically introduces error. However, the *magnitude* and *implications* of this variance in practice were previously unquantified. In empirical science, knowing that a measurement is noisy (such as here) is different from knowing the extent of that noisiness. Our work focuses on shifting the field's unquantified expectation of having "some variance" to rigorously measuring it. Here, we show that this variance is substantial enough to yield structurally disjoint circuits for the same task, which challenges the reliability of claims made based on single-run circuit discovery (as is common practice in foundational papers such as Wang et al., 2023 and the original EAP/EAP-IG papers). Knowing that a measurement is noisy is different from demonstrating that the noise is large enough to undermine scientific conclusions.
> - On the sample size and variance: We thank the reviewer for this point and have clarified the paper. Our bootstrap analysis explicitly maintains a fixed sample size while resampling with replacement from the original dataset. The high variance that we observe is therefore not due to a lack of data, but due to the method's sensitivity to the specific *composition* of the dataset, which highlights a form of overfitting to the discovery sample.
> - On the scope and focus on EAP-IG: We chose EAP-IG because it represents the current state-of-the-art for efficient, gradient-based edge discovery. While other methods (e.g., ACDC) are powerful, they are computationally much more expensive, making the large-scale statistical analysis performed here (thousands of runs) prohibitive. We see our contribution not as an analysis of all methods, but as establishing a **statistical estimation framework** that is essential to the field and can be applied in future work to Path Patching, Edge Pruning, or SAE-based circuit discovery.
>
> Answers to questions:
> - On why bootstrap yields higher variance: EAP-IG appears to overfit to artifacts within a specific sample $D$. When bootstrapping, we perturb the specific instances included, which may cause the method to catch different noise artifacts. In contrast, meta-distribution shifts (paraphrasing) may preserve the semantic concept of the task more robustly than the specific noise correlations of a fixed dataset.
> - On disjoint circuits (Table 2): The fact that two disjoint circuits yield similar error rates (0.20) suggests non-identifiability more strongly than the spuriousness of said circuit, considering the rather low value of said error rate. If circuits were systematically identifiable, then as the error rate of a family of circuits goes to zero, one would expect their paired Jaccard overlap to approach 1. The low value observed here suggests that this is not the case.
>
> We have systematically incorporated the suggested corrections regarding citations and fixed the presentation issues (definitions, figure captions) in the new version of the manuscript. We hope that we have addressed the reviewer's concerns and look forward to further discussion.

---

> > ### Comment · Reviewer_6Yb9 · 2025-11-24
> >
> > Thank you to the authors for the thoughtful rebuttal. Although the authors clarified several points and corrected presentation issues, these changes do not resolve the central scientific concerns. The main finding, that EAP-IG circuits exhibit high variance, remains largely unsurprising given the method’s reliance on first-order approximations, and the rebuttal does not provide substantial new insight beyond reframing this as a measurement exercise. While the authors mention that the high variance is not due to a lack of data, they do not report sample size details. They neither justify whether the sample size is large enough for stability estimation nor analyze the sensitivity of EAP-IG variance to sample count.
> >
> > The narrow scope of the empirical analysis is also unchanged. Focusing exclusively on EAP-IG may be practical, but without even a limited comparison to alternative circuit-discovery methods, the generality of the conclusions is unclear. Additionally, the explanations offered for key phenomena, such as higher bootstrap variance and structurally disjoint circuits with similar errors, are speculative and insufficiently supported by evidence. Overall, while the paper is now cleaner and better presented, the core issues related to novelty, methodological depth, and generality remain unresolved. Therefore, I’m keeping my original score.

---

> > > ### Author Response · Authors · 2025-11-26
> > >
> > > We thank the reviewer for their continued engagement.
> > >
> > > While we understand the reviewer's perspective that variance is expected from first-order approximations, we respectfully maintain that in empirical science, the *magnitude* of that variance is what determines the utility of a tool. Knowing a measurement is noisy is different from knowing that the noise is so high that the method returns structurally disjoint circuits for the same input. The latter renders the tool unreliable for scientific discovery without the statistical controls we propose. Our work provides the concrete evidence necessary to measure the noise to prevent false conclusions, rather than simply assuming the existence of said noise.
> > >
> > > **On sample size:** We apologize if our this was unclear in the previous response. We have clarified in Section 3.3 that for bootstrap resampling, we resample $n=100$ datasets from the original dataset $D$, with each dataset containing $|D|/5$ points (with replacement). $|D|$ ranges from 500 to 1000 in the datasets we use.
> > >
> > > Regarding the disjoint circuits with similar errors, we interpret this as a set of circuits that perform approximately equally well but rely on different features. If EAP-IG arbitrarily selects one disjoint circuit over another due to slight noise, it indicates the method is underspecified for the task. This is not speculation, but a direct interpretation of the Jaccard/Error data.
> > >
> > > We hope this clarifies our position for the final decision.

---

### Comment · Area_Chair_TBy9 · 2025-11-25
**Please discuss**

Several reviewers have not responded to the authors' rebuttals. Please read and respond to them. Have the rebuttals addressed your concerns or clarified anything?

---

### Meta-Review · Area_Chair_8Sac · 2025-12-09

**Summary:**

This study proposes a framework wherein mechanistic interpretability—and more specifically, the causal graph/circuit discovery subset of the field—is viewed as statistical estimation. Under this framework, the variance and robustness of edge attribution patching with integrated gradients (EAP-IG; a common method in circuit discovery), is empirically investigated. It is observed that EAP-IG has high variance and is sensitive to hyperparameters. The authors use this evidence to call for improved reporting of stability metrics.

While this study investigates a topic of interest to many in the field, reviewers agreed that the experimental setup could use some refining. A common concern is that EAP-IG does not appear to be well-calibrated in the experiments. Another concern is that the scope of experimentation is narrow: there exist other common circuit discovery techniques that are not investigated, so it is still unclear how general this problem truly is. While the contribution is more of a scientific framework than an empirical one, the evidence in favor of this framework's utility is still fuzzy.

**Reviewer Concerns:**

On the validity of the EAP-IG implementation and experimental setup: Reviewers brought up many concerns on this point.

* Reviewer 6Yb9 mentions small sample sizes and the approximate nature of the method. Responses have clarified that there are statistical controls in place (which the reviewer seems satisfied with), but this does not address the related concern on scope (which follows).

* Reviewer EUfV mentions issues relating to the use of discrete metrics, the application of stability metrics in light of there potentially not existing identifiable circuits for a given task, and high error rates. The discussion led to clarifications, but these do not appear to have been completely addressed.

* Reviewer 4NAY mentions non-standard uses of the greedy circuit search algorithm. This has been discussed in detail, but I do not think it has been sufficiently addressed. It is, in fact, more common to search over circuit sizes and report faithfulnesses across many sizes. While one does generally aim to achieve the best trade-off between minimality and faithfulness, connectivity has been treated as more of a necessary (rather than sufficient) condition in prior work.

On the scope of experimentation: there now exist other circuit discovery techniques, such as information flow routes, uniform gradient sampling, and (for smaller models) exact activation patching that could have been analyzed in a similar manner. I think that the statistical foundations of the paper are sound, but the breadth of evidence is somewhat lacking.

While each of these concerns in isolation appear minor, that each reviewer pointed out some flaw in the experimental setup acts as evidence that the paper's arguments could be significantly strengthened via a thorough pass over the experimental setup and/or its presentation.

**Reviewer Scores:**

Given the discussions, the reviewers did not appear on track to significantly change their scores. The follow-ups to which the authors did not have a chance to respond would likely not have swayed me to change my scores.

---

### Decision · Program_Chairs · 2026-01-26

Reject